# Phylogenetic structure of moth communities (Geometridae, Lepidoptera) along a complete rainforest elevational gradient in Papua New Guinea

**Sentiko Ibalim**[1,2]*, **Pagi S. Toko**[1,2,3], **Simon T. Segar**[4], **Katayo Sagata**[5], **Bonny Koane**[3], **Scott E. Miller**[6], **Vojtech Novotny**[1,2], **Milan Janda**[2,7,8]

1 Faculty of Science, University of South Bohemia, Ceske Budejovice, Czech Republic, 2 Institute of Entomology, Biology Centre, Czech Academy of Sciences, Ceske Budejovice, Czech Republic, 3 New Guinea Binatang Research Centre, Madang, Papua New Guinea, 4 Department of Crop and Environment Sciences, Harper Adams University, Newport, United Kingdom, 5 PNG Institute of Biological Research, Madang, Papua New Guinea, 6 Department of Entomology, Smithsonian Institution, National Museum of Natural History, Washington, DC, United States of America, 7 Escuela Nacional de Estudios Superiores Unidad Morelia, Universidad Nacional Autónoma de México, Morelia, Michoacán, México, 8 Faculty of Science, Department of Zoology, Palacky University Olomouc, Olomouc, Czech Republic

* sentikoibalim@gmail.com

**Data Availability Statement:** The data underlying the results presented in the study are available

## Abstract

We use community phylogenetics to elucidate the community assembly mechanisms for Geometridae moths (Lepidoptera) collected along a complete rainforest elevational gradient (200–3700 m a.s.l) on Mount Wilhelm in Papua New Guinea. A constrained phylogeny based on COI barcodes for 604 species was used to analyse 1390 species x elevation occurrences at eight elevational sites separated by 500 m elevation increments. We obtained Nearest Relatedness Index (NRI), Nearest Taxon Index (NTI) and Standardised Effect Size of Faith's Phylogenetic Diversity (SES.PD) and regressed these on temperature, plant species richness and predator abundance as key abiotic and biotic predictors. We also quantified beta diversity in the moth communities between elevations using the Phylogenetic Sorensen index. Overall, geometrid communities exhibited phylogenetic clustering, suggesting environmental filters, particularly at higher elevations at and above 2200 m a.s.l and no evidence of overdispersion. NRI, NTI and SES.PD showed no consistent trends with elevation or the studied biotic and abiotic variables. Change in community structure was driven by turnover of phylogenetic beta-diversity, except for the highest 2700–3200 m elevations, which were characterised by nested subsets of lower elevation communities. Overall, the elevational signal of geometrid phylogeny was weak-moderate. Additional insect community phylogeny studies are needed to understand this pattern.

## 1 Introduction

Understanding how ecological communities are assembled and maintained is one of the main themes of community ecology [1–3], becoming increasingly important also practically as

**Funding:** The study was supported by the Praemium Academiae to VN Czech Science Foundation (23-07776S), The European Research Council (669609) and GAJU n. 014/2022/P supporting SI and by CONAHCYT program Investigadores por Mexico, project No. 338. 338 to SEM.

**Competing interests:** All authors have declared that no competing interest exist.

communities have to respond to the rising intensity of environmental change [4]. Evolutionary processes and dispersal along large-scale environmental gradients, such as latitudinal, elevational or rainfall gradients, generate regional species pools which are then filtered by local environmental factors and biotic interactions to form local assemblages [1,2,5–7]. Elevation represents one of the key gradients of rapid ecological change over short spatial scales [8]. Elevational gradients therefore vary across geographic regions, shaped by their latitudinal position, elevation range, geological histories, and topologies [9]. The linear decrease of temperature with increasing elevation, as well as more complex patterns in solar radiation, air humidity, rainfall and soil nutrients lead to high turnover of species with elevation, generating in turn elevation gradients of biotic interactions including competition, predation, and parasitism [9–11]. The community assembly process along elevation gradients are therefore entities shaped by numerous abiotic and biotic drivers [9,12,13]. Further, elevation gradients act as generators of species on evolutionary time scales [14].

It has been long recognized that in order to understand the assembly of ecological communities we need to incorporate the analysis of phylogenetic relationships of their constituent species [15]. However, this approach has only become feasible in the last couple decades, with increasing accessibility of molecular data and comprehensive taxon-level and community-level phylogenies [16,17]. Community phylogenetics opens new ways to examine the historical and contemporary processes driving community function and structure [16–19] but see [20]. Environmental filtering selects species with similar traits and since such species tend to be closely related, this process leads to a clustered distribution of species on wider phylogeny [16]. In contrast, biotic interactions favour coexistence of species with dissimilar traits that minimise competition [1,17,18]. Competition therefore acts to produce phylogenetically overdispersed communities [1,17,18]. In some scenarios, competitive exclusion could apply to entire clades, leading thus also to a phylogenetically clustered community [20,21]. Further, the pattern of phylogenetic diversity produced by environmental filtering and competition also depends on phylogenetic conservatism of individual species traits [22,23].

Geometrid moths (Geometridae, Lepidoptera) represent a highly diverse taxon of insect herbivores, comprising ~24,000 described species [24]. They are widely used as a model group for community studies and biodiversity monitoring [25,26] because of their high abundance, species diversity, wide geographic distribution, wide range of host specialisation, sensitivity to environmental changes, and relatively good taxonomic knowledge [27]. Geometridae species richness and distribution have been widely studied, including the patterns along elevational gradients [13]. A comprehensive analysis of elevational gradients documented the prevailing pattern of mid-elevation species diversity maximum for geometrids [28–30]. However, these studies failed to identify a well-defined set of drivers responsible for the mid-elevation maximum, although plant primary productivity, together with decreasing land area and predation intensity with increasing elevation tend to be influential.

Only a few of the geometrid studies have so far included phylogenetic relationships on the community level [31]. The geometrid communities in the Andes became more phylogenetically clustered with increasing elevation, potentially indicating a growing role of environmental filtering [31]. A similar pattern of increasing phylogenetic clustering with increasing elevation was documented in ant communities from the temperate zone [32]. Here we assess the community phylogenetic structure of eight geometrid moth communities along a complete rainforest elevation gradient spanning from the lowlands at 200 m a.s.l to the timberline at 3700 m a.s.l along the Mt. Wilhelm transect in Papua New Guinea (PNG).

We aim to test the hypothesis that the underlying effect of interspecific competition is predominant in community composition at low elevation and that the impact of environmental filtering becomes increasingly important with increasing elevation. This would result in the

transition of community phylogeny from over-dispersion in the lowlands to clustering at high elevations. Further, the increasing phylogenetic clustering with elevation combined with the mid-elevation maximum in species diversity [13,33] could lead to a range of trends in community phylogenetic diversity, from monotonic decrease with elevation to a mid-elevation maximum, not necessarily coinciding with the maximum in species diversity. Further, this study aims to create baseline data on the elevation distribution of a taxonomically well-defined and species rich taxon for the future monitoring of the climate change impact on the rainforest biodiversity of PNG.

## 2 Methods

### 2.1 Community composition and taxonomy

The geometrid moth communities were sampled by light trapping along an elevational gradient on the slope of Mt. Wilhelm (Fig 1). Mt Wilhelm is the highest peak (4509 m a.s.l) of PNG, comprising an uninterrupted gradient of tropical rainforests from lowlands to the timberline at 3700 m a.s.l. Eight communities were sampled between 200 to 3700 m a.s.l at 500 m elevational increments (Fig 1). For further details on sampling sites and their environmental conditions see Sam et al. [34,35] and Toko et al. [29].

The surveys occurred from May to October 2009 and from December to January 2010 in order to cover both wet and dry seasons [29,31]. At each elevation, moths were attracted to a 1 x 2 m white sheet positioned in primary rainforest using a 240W Mercury-vapour light operated from 18:00 to 24:00 for 7–10 trapping nights per elevation [29]. The moths were hand collected and sorted to morphospecies. In total, 16,424 individuals comprising 1,102 species were collected and used by Toko et al. [29] for ecological trend analyses (Table 1). From these, a total of 1,582 representative specimens were barcoded using the standardised protocol of the Biodiversity Institute of Ontario [37]. These specimens represented combinations of individual morphospecies present at each elevation. The barcode sequence data produced 1,390

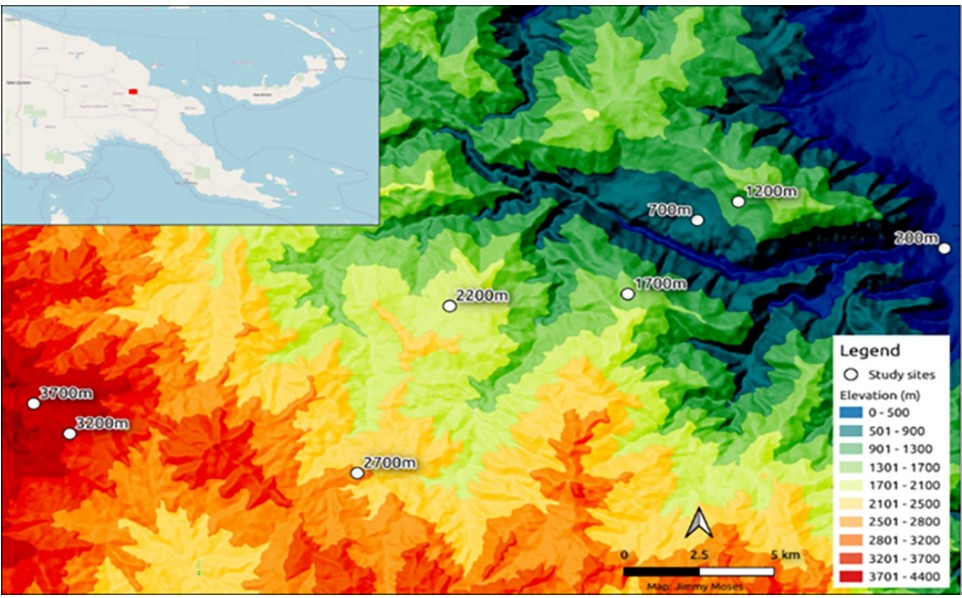

**Fig 1. Wilhelm elevational gradient with eight study sites at 500 m elevational increments from 200 to 3700 m a.s.l (blue to red). The Mt.** Insert: Map of Papua New Guinea with Mt. Wilhelm study area shown in red square. Reprinted from Moses et al. [36] under a CC BY licence, with permission from Jimmy Moses, original copyright 2021.

**Table 1. Abundance and species diversity of all species and the sequenced species along the Mt. Wilhelm elevational gradient.**

| Study site | Elevation (m a.s.l) | Toko et al. [29] Abundance (counts) | Morpho species | This study Sequenced species per elevation |
|---|---|---|---|---|
| Kausi | 200 | 2,311 | 201 | 179 |
| Bobvrai | 700 | 1,805 | 239 | 198 |
| Koviamarai | 1200 | 2,194 | 391 | 283 |
| Bananumbu | 1700 | 3,903 | 403 | 289 |
| Sinopass | 2200 | 2,631 | 305 | 232 |
| Kiagimanigi | 2700 | 1,134 | 197 | 156 |
| Kombugo Manuno | 3200 | 2,091 | 182 | 45 |
| Piunde Lake | 3700 | 355 | 37 | 8 |
| **No. species** | | **16,424** | **1,102** | **604** |

species occurrences along eight elevational communities (from here on 'our dataset') used in this study after excluding low-quality sequences (Tables 1 and S1). These sequences consisted of 604 molecular species referred to as Barcode Index Numbers (BINs), based on an algorithm similar to a 2.2% divergence threshold [38] used for the analysis of phylogenetic structure. The 1,390 occurrences of the 604 species were used to generate a presence/absence matrix of species x elevations where the elevational distribution of each species was interpolated to a single missing elevation flanked by both the nearest lower and higher elevations where the species was recorded. The presence/absence matrix can be found in S1 File. The sequence data are publicly available in the Barcode of Life database (BOLD, data set PAGIB) (https://www.boldsystems.org/index.php), and the specimens are deposited at the Binatang Research Centre, PNG, with duplicates at the National Museum of Natural History, Smithsonian Institution, USA. Further, we made all datasets and R-Scripts used for the analyses freely accessible in the Zenodo repository here: (https://zenodo.org/records/12551947).

## 2.2 Community phylogenetic structure

We constrained the phylogeny of our 604 species to the phylogeny generated by Murillo-Ramos et al. [39]. The OTU names of our sequences adopted the names assigned in BOLD under the dataset name DS-PAGIB as taxonomically identified by Scott E Miller. For the sequences missing genera or other OTU names, we built a Maximum Likelihood super-tree combining our 604 sequences and the 1,206 species level identified sequences from Murillo-Ramos et al. [39] and assigned the OTU names based on where they branched among Murillo-Ramos species. This generated a tree resolved to genus level, which simplified the analysis as well as avoided polytomies which have been found to reduce the power of phylogenetic matrices to detect phylogenetic structure [40]. However, we acknowledge that this resulted in assignments of some PNG species to generic placements, that, while taxonomically close, may not be correct, so they should not be considered for taxonomic identifications.

Three outgroup members were used and include *Coronidia orithea* (Sematuridae), *Pseudobiston pinratanai* (Pseudobistonidae) and *Chrysiridia ripheus* (Uraniidae) all of which are suitable outgroups of the Geometridae moth family [41] as used in Sihvonen et al. [42]. Bayesian priors were set in BEAUti tool in BEAST V2.3.1 to run for 20M generations sampling every 5000 generations with a 10% burn-in and molecular clock set as relaxed clock log normal with a substitution rate of 0.1207 as identified in IQtree model selection. The sequence matrix was partitioned by first plus second and third (1+2) +3) codon positions and GTR+F+I+G4 as the

best substitution model as selected in IQtree: http://iqtree.cibiv.univie.ac.at/ [43] based on the best model selection under jModelTest with the "–m TEST" option (S2 Table). To keep the phylogeny resolved and bifurcating as possible [44], the tree priors were constrained at three levels that reflect subfamilial, tribal and generic relationships by keeping each level monophyletic following OTU classifications of Geometridae moth group in Murillo-Ramos et al. [39]. A uniform birth death model was selected. The trees were retrieved in BEAST2 [45,46] on XSEDE within Cyber infrastructure for phylogenetic research (CIPRES: https://www.phylo.org/). The convergence of the trees was assessed in Tracer V1.6: all parameters have effective sample size (ESS) >200 [46] with the ESS of posterior probability at 845. The 3,600 trees generated were annotated to obtain the consensus tree using the Tree Annotator tool in BEAST V2.3.1 (S2 and S3 Files). To assess branch support, we performed 1,000 replicates of the ultrafast bootstrap approximation (UFB) [47,48] and 1,000 replicates of the branch-based, Shimodaira–Hasegawa approximate likelihood ratio test [49] in IQtree. The three outgroup members were removed to retain the 604-genus level identified and constrained tree for subsequent analysis. The constrained tree can be found as a nexus file (S2 File) and an example phylogenetic tree with node support can be found in S3 File and the raw data information in S1 Table.

We based assessment of community phylogenetic structure on (i) Net Relatedness Index (NRI) measured as the pairwise phylogenetic distance across all individuals in the entire tree and (ii) Nearest Taxon Index (NTI) measuring phylogenetic distance to the nearest neighbour species [16]. Both are measures of the differences between the observed and randomly generated null communities [16,50]. In our case, both indices quantify the distributions of moth species at each elevational site relative to the gradient-wide occurrences of the moths. The species x elevation occurrence matrix (N = 13,90) was randomised on the Bayesian tree (N = 604) by shuffling taxon labels using the function *cophenetic.phylo* in R package *picante* [16,32,51,52]. The mean pairwise distance was then used to calculate NRI and NTI using the function *as. matrix*. The *picante* output result for the community phylogenetic dispersion for these analyses are reported in S3 Table. The positive values of NRI and NTI indicate phylogenetic clustering and the negative values indicate phylogenetic over-dispersion within a community [16,17,53]. The two indices were not correlated (Pearson's r = 0.165, P = 0.696), we therefore report both of them [31].

We also used Faith's Phylogenetic diversity (PD) as the sum of the branch length in a phylogeny connecting all species in a community to characterise each community as obtained by the function *pd* [54,55]. However, PD is dependent on the number of species included in the community [56]. To correct this, we also used standardised effect size of Faith's Phylogenetic Diversity (SES.PD) measuring phylogenetic diversity for uneven sample sizes [53]. It is the difference between the observed and randomly generated Faith's PD values standardised by the standard deviation of the randomised PD [16,50]. The phylogenetic dispersions obtained were visualised using the ggplot function *ggplot2*. The correlations of three phylogenetic indices and elevation were analysed using function *pair.panels* in R package *psych*. All analyses were performed using the R-package *picante* [50,57]. R scripts are available in Zenodo public repository here: (https://zenodo.org/records/12551947).

## 2.3 Environmental predictors of community phylogenetic structure

The community phylogenetic structure, quantified by the NRI, NTI, and SES.PD indices, were tested against three potential predictors: mean annual temperature as a key abiotic factor, plant species richness quantifying potential resources, and the predation pressure as a key top-down control factor using linear regression models. We limited the analysis to these variables

**Table 2. The phylogenetic parameters of geometrid communities and the ecological and environmental variables used to explain them.** The values of plant richness, predator abundance and temperature were standardised to z-scores and their raw data can be found in S1 Table.

| Localities | | Response variables | | | Predictors | | |
|---|---|---|---|---|---|---|---|
| Locality | Elevation | NRI | NTI | Faith's PD | Plant species richness | Predator abundance | Mean temperature |
| Kausi | 200 | 0.21 | 2.51 | -2.22 | -0.22 | 1.51 | 1.48 |
| Bobvrai | 700 | -0.08 | 1.21 | -0.88 | 1.11 | 1.19 | 1.08 |
| Koviamarai | 1200 | -0.31 | -0.35 | 0.69 | 1.26 | 0.53 | 0.56 |
| Bananumbu | 1700 | 2.81 | 1.32 | -1.35 | 0.54 | -0.11 | 0.36 |
| Sinopass | 2200 | 3.05 | 2.11 | -2.59 | -0.84 | -0.43 | -0.14 |
| Kiagimanigi | 2700 | 1.66 | 4.60 | -5.43 | 0.29 | -0.05 | -0.59 |
| Kombugo Manuno | 3200 | 0.10 | 4.61 | -4.43 | -0.12 | -0.87 | -1.19 |
| Piunde Lake | 3700 | 2.63 | 2.18 | -3.11 | -2.01 | -1.76 | -1.55 |

to avoid model overfitting as there were only eight data points. Mean annual temperature decreases at the rate of 0.54˚C per 100 m elevation from 27.4˚C in the lowlands (200 m a.s.l) to 8.37˚C at the timberline (3700 m a.s.l) along the Mt. Wilhelm transect, based on one-year measurements by data loggers [34]. Vegetation diversity was measured as the number of woody species with DBH ≥ 5 cm in three 20 x 20 m primary forest plots at each elevation [29] although we acknowledged that this may have precluded information for those species specialised in lower plant forms or plant litter [58]. The abundance of predators was derived by Toko et al. [29] as an index combining the data on the abundance of insectivorous bats [59], insectivorous birds, and ants [60]. Bats were monitored by audio surveys, birds by point counts and ants collected at tuna baits. Bats are a major group of predators of adult moths [61], while birds and ants are common predators of caterpillars [62]. Each taxon of predators was given an equal weight. For statistical convenience the abundance values of these predictors were standardised to Z-scores (Table 2) prior to the linear regression analyses.

The correlations were performed using *lm* function for each variable separately plus their combinations in R version 4.2.1. The models obtained were assessed using delta Akaike information criterion (ΔAICc) (Table 3). Those models with ΔAIC values <2 or nearest are considered as plausible explanations for the regressions [63]. Because the diversity pattern for geometrid moths on elevational gradients is often nonlinear [13,29,64], we also assessed second order polynomials of the explanatory variables using poly function (S4 Table) [57].

## 2.4 Community phylogenetic dissimilarity

The community phylogenetic dissimilarity of Geometridae moths was assessed using Phylogenetic Sorensen index (PhyloSor), an extension of Sorensen index that incorporates phylogenetic relationships to measure phylogenetic dissimilarity among communities [65]. It is calculated as the sum of the phylogenetic distances between species that are shared between the two communities, divided by the total phylogenetic distance in both communities [55,65,66]. Low values indicate that the two communities are composed from closely related species and high values indicate pairs of communities that are distantly related [55]. We used the PhyloSor function in R package *picante* for this analysis. PhyloSor was further decomposed into net PhyloSor, nestedness and species turnover [65,67,68] using the function *phylo.beta.pair* in R package *betapart*. The distance matrices obtained were visualised as heatmaps for the net (Fig 4B) and for both nestedness (S2A Fig) and turnover (S2B Fig) using *heatmaply* function. We also generated phylogeny-free Sorensen dissimilarity of the moths based on presence/absence of the moths to assess the differences in Sorensen and phylogenetic Sorensen

**Table 3. The linear regression models (*lm*) for the effect of plant species richness, temperature, and predator abundance on Geometridae phylogenetic structure.** The models are ranked by ΔAICc from good to worst (top to bottom). Models with ΔAICc <2 or nearest (bolded) are equally supported.

| Dependent | Model | LogLik | AICc | ΔAICc | Weight |
|---|---|---|---|---|---|
| NRI | Null | -13.688 | 33.8 | 0.00 | 0.540 |
| | Plant species richness | -11.944 | 35.9 | 2.11 | 0.188 |
| | Predator abundance | -11.951 | 35.9 | 2.12 | 0.187 |
| | Temperature | -12.898 | 37.8 | 4.02 | 0.072 |
| | Predator abundance + Temperature | -10.332 | 42.0 | 8.22 | 0.009 |
| | Plant species richness + Predator abundance | -11.533 | 44.4 | 10.62 | 0.003 |
| | Plants species richness + Temperature | -11.915 | 45.2 | 11.39 | 0.002 |
| | Plants species richness + Temperature + Predator abundance | -10.047 | 60.1 | 26.32 | 0.000 |
| NTI | Null | -14.991 | 36.4 | 0.00 | 0.694 |
| | Temperature | -13.705 | 39.4 | 3.03 | 0.153 |
| | Plant species richness | -14.486 | 41.0 | 4.59 | 0.07 |
| | Predator abundance | -14.544 | 41.1 | 4.71 | 0.066 |
| | Predator abundance + Temperature | -11.319 | 44.0 | 7.59 | 0.016 |
| | Plants species richness + Temperature | -13.702 | 48.7 | 12.36 | 0.001 |
| | Plant species richness + Predator abundance | -14.421 | 50.2 | 13.79 | 0.001 |
| | Plants species richness + Temperature + Predator abundance | -10.324 | 60.6 | 24.27 | 0.000 |
| SES.PD | Null | -16.191 | 38.8 | 0.00 | 0.535 |
| | **Temperature** | **-14.095** | **40.2** | **1.41** | **0.265** |
| | Plant species richness | -15.133 | 42.3 | 3.48 | 0.094 |
| | Predator abundance | -15.212 | 42.4 | 3.64 | 0.087 |
| | Predator abundance + Temperature | -12.272 | 45.9 | 7.10 | 0.015 |
| | Plants species richness + Temperature | -14.014 | 49.4 | 10.58 | 0.003 |
| | Plant species richness + Predator abundance | -14.956 | 51.2 | 12.46 | 0.001 |
| | Plants species richness + Temperature + Predator abundance | -10.792 | 61.6 | 22.80 | 0.000 |

Plant Sorensen dissimilarity distance was significantly positively correlated with moth phylogenetic dissimilarity (r = 0.4231, Mantel test, P<0.05) which suggested parallel changes in beta diversities between geometrids and their food resources.

dissimilarities with increasing elevation (Fig 4A). A hierarchical cluster dendrogram was constructed using *hcluster* function to explore community similarity using phylogenetic pairwise distances between communities as generated using *comdist* function in R package *picante* (S4 Fig and S4 File) [57]. In addition, we assessed the effect of plant Sorensen dissimilarity on Geometridae moth phylogenetic Sorensen dissimilarity distribution using the Mantel test. This analysis helps to determine how differences in plant species and their compositional differences predict differences in moth evolutionary diversity along the gradient.

## 3 Results

### 3.1 Species and phylogenetic diversity trends

We reconstructed a phylogenetic relationship for 604 BINs used here as OTUs, in Geometridae, representing six subfamilies, 45 tribes and 153 genera. These species were recorded in 1390 species x elevation occurrences along Mt. Wilhelm elevational gradient. The subfamily level composition was following: Desmobathrinae (10 species, 1.7% of total species), Ennominae (226, 37.4%), Geometrinae (211, 34.9%), Larentiinae (95, 15.7%), Oenochrominae (10, 1.7%) and Sterrhinae (52, 8.6%). Our dataset covered 55% of the total of 1,102 geometrid species recorded along the gradient [29]. The species richness and Faith's PD were positively

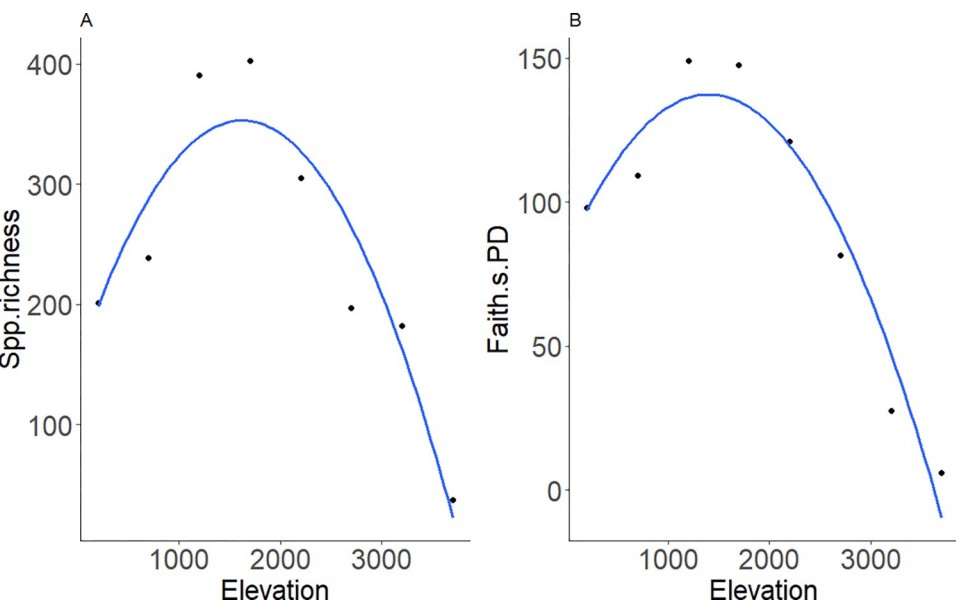

**Fig 2.** Species richness (A) and Faith's phylogenetic diversity (B) of Geometridae moth communities along Mt. Wilhelm elevational gradient. Y-axis are observed species richness (A) and Faith's PD (B) both fitted with second order linear model function with elevation as an explanatory variable.

correlated (r = 0.928, p <0.001) and both exhibited a mid-elevation maximum between 1200 and 1700 m a.s.l (Fig 2A and 2B).

## 3.2 Community phylogenetic diversity

Both NRI and NTI values vary markedly along the elevation gradient exhibiting random and non-random patterns at different parts of the gradient. NRI indicated phylogenetic clustering in 1700, 2200 and 3700 m sites with the remaining sites not significantly different from random distribution (Fig 3A). NTI on the other hand showed significant clustering at ≥2200 m elevations with other sites mostly around mid-elevation exhibiting random patterns (Fig 3B). There was no evidence of phylogenetic overdispersion in both NRI and NTI in any of the communities. The SES.PD metric exhibited an opposite but strongly correlated trend to that of NTI indicating lower phylogenetic diversity towards higher elevations between 2200–3700 m a.s.l (Fig 3C). Hence, all metrics were consistent with an increase in phylogenetic clustering at and above 2200 m a.s.l.

Multiple correlation matrices among elevation and the three phylogenetic variables (NRI, NTI and SES.PD) indicated a significant negative correlation only between SES.PD and NTI (r = 0.97, P <0.0001) (S1 Fig).

## 3.3 Phylogenetic β-diversity

The β-diversity among adjacent communities increased with increasing elevation, both for non-phylogenetic (Fig 4A) and phylogenetic (Fig 4B) beta diversity indices with higher phylogenetic dissimilarity among geographically distant elevation sites.

Beta diversity was dominated by species turnover (S3 Fig), but nestedness became prominent at higher elevations (S2B Fig) starting from the 2200 m a.s.l site. The cluster analysis formed two principal clusters, one highland cluster formed by top two elevations (3200 and 3700 m a.s.l) and the other by the remaining sites which further splits into mid (1700–2700 m a.s.l) and lowland (200–1200 m a.s.l) elevation clusters (S4 Fig).

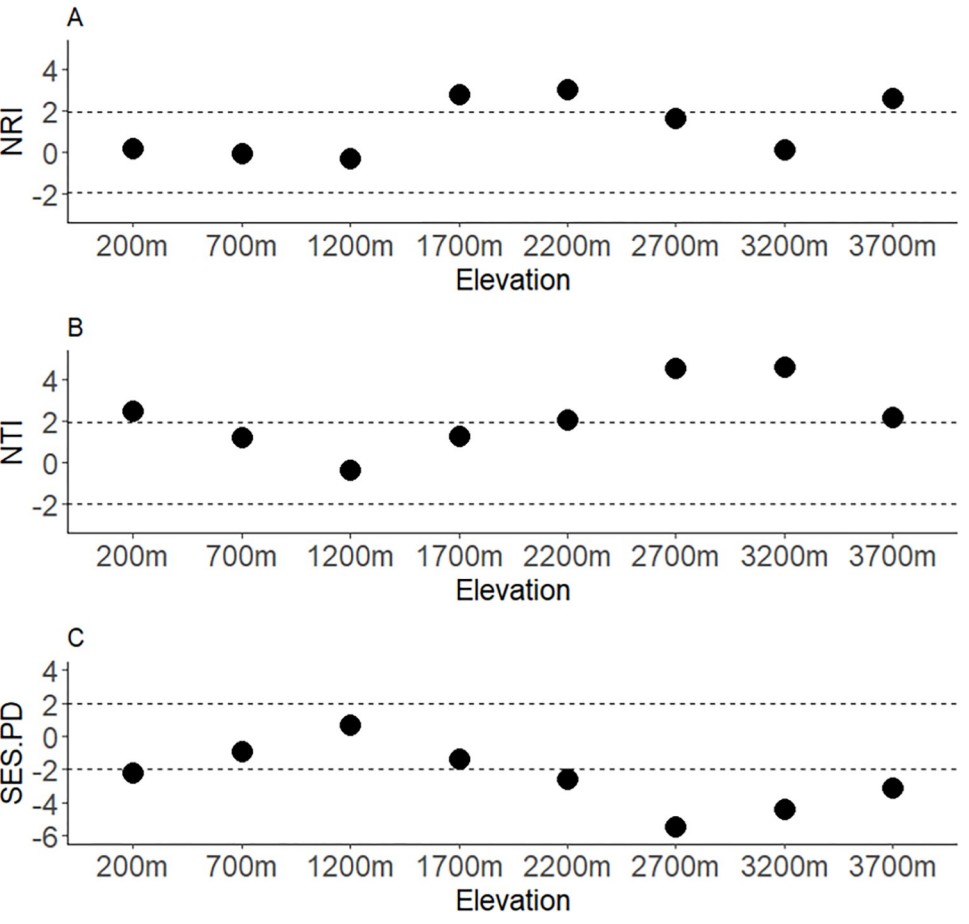

**Fig 3.** Phylogenetic structure of geometrid moths on Mt. Wilhelm elevational gradient based on NRI (A), NTI (B) and SES.PD (C). In each plot, the horizontal dashed lines at 1.96 and -1.96 demarcated area of randomly distributed communities while the sites ≥1.96 show phylogenetic clustering (NRI, NTI) or low phylogenetic diversity (SES.PD) and the sites ≤-1.96-line indicated phylogenetic over-dispersion (NRI, NTI) or high phylogenetic diversity (SES.PD). NRI = Nearest relatedness index, NTI = Nearest taxon index, SES.PD = Standardised effect size of Faith's PD.

## 3.4 Biotic and abiotic predictors on phylogenetic diversity

There was no significant correlation between the community phylogenetic structure measured by SES.PD, NRI and NTI and the three explanatory variables: mean annual temperature, plant diversity and predator abundance (Table 3). Only temperature indicated a marginal linear effect on phylogenetic diversity (P = 0.08) (Table 3) with ΔAIC distance less than two [63]. Second order polynomial regressions marginally only supported quadratic relationship in NTI and plant species richness (P = 0.092) (S4 Table). It is notable that the models with single predictor variables have tendencies to exhibit better predictive effects compared to the more complex models (Table 3).

## 4 Discussion

Our study added a phylogenetic component to the analysis of highly diverse insect communities along an extensive rainforest elevation gradient [29]. The goal was to evaluate the phylogenetic structure and use this to assess the mechanism influencing community assembly. Such studies are uncommon in highly diverse tropical forests, where insect communities comprise

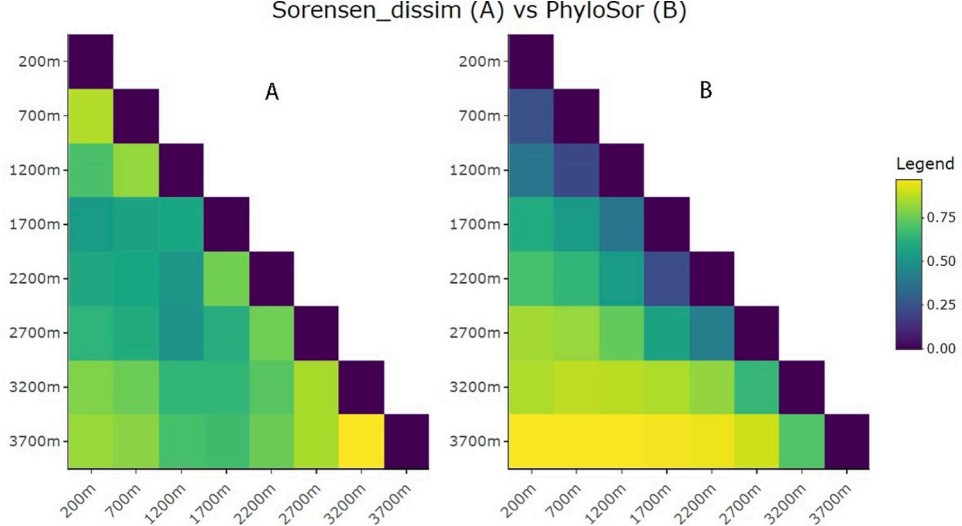

**Fig 4.** Sorensen measure of community dissimilarity (A) and Phylogenetic Sorensen beta-diversity (B) among the eight geometrid communities on Mt. Wilhelm transect. The values range from 0 for highly similar (blue squares) to 1 for dissimilar (yellow squares) paired communities.

exceptionally high numbers of species [69,70], and where unresolved taxonomies and unknown phylogenies are inevitable. Using the well-known model taxon, the geometrid moths, we bridged this gap by combining large-scale global phylogenies [39] with extensive regional barcode data to explore the possible mechanisms. The distribution of NRI and NTI showed distinctive patterns, with a tendency to cluster at higher elevations with no evidence of overdispersion. The clustering at higher elevations (in NTI), the strong correlation between NTI and phylogenetic diversity (SES.PD) and the marginally significant correlation between temperature and phylogenetic diversity suggested that temperature is influential in driving the co-occurrence of genetically closely related individuals at higher elevation. This pattern is consistent with beta-diversity increasing with elevation, showing gradual turnover between adjacent lowland sites with a more abrupt change among spatially distant communities. Communities also become increasingly nested at sites 2700 m a.s.l and higher. The genetic data also divided the moth communities along the transect into highland communities formed by the top two elevations (3200 and 3700 m a.s.l) and lowland ones formed by rest of the sites which further clustered into mid (2700, 2200, and 1700 m a.s.l) and lowland (1200, 700 and 200 m a.s.l) groups, even though Mt. Wilhelm is a continuous forested gradient, reflecting the similarity in species composition. There was a lack of strong correlation between biotic variables (plant species richness and predator abundance) and any of the phylogenetic matrices, although plant species richness showed a quadratic effect on NTI, indicating its effect on moth communities at community level as well as their correlated higher diversities at mid elevation [29].

## 4.1 Faunal composition and phylogenetic structure

The phylogenetic diversity and species richness of geometrid moths reached their peak at mid-elevation (1200 and 1700 m a.s.l) along Mt. Wilhelm. Although these metrics are frequently congruent [54,55,71] the pattern may be influenced by various ecological and evolutionary factors that impact community composition over time [72,73]. Further, this may be attributed to

the diversity and distribution patterns of subfamily level OTU [13,28,31,74]. For example, on Mt. Wilhem, the number of species decreased with elevation in Sterrhinae and Geometrinae, increased in Larentiinae, and remained constant for Ennominae [29]. This resulted in particularly higher overlap in subfamilies and their associated species numbers at mid-elevation which may have influenced the mid peak diversity patterns observed. A consistent similar pattern of subfamily compositional diversity at mid-elevation within this moth group was reported across other elevational gradients in other biogeographic regions [13,27,28,31,56,74,75].

Phylogenetic clustering was observed by both NRI and NTI along a portion of the gradient, but there was no evidence of phylogenetic overdispersion. Further, we detected non monotonous trends in phylogenetic parameters of geometrid communities with increasing elevation. This contrasted to the geometrid communities in the Andes, where NTI values increased monotonically towards higher elevations [31]. In contrast, at lower elevations, many taxa including insects [31,32], plant communities [71] and hummingbirds [76] were shown to exhibit overdispersion. This is associated with various factors including competition, species interactions, resource availability and habitat heterogeneity [76]. Brehm and Fiedler [31] observed that geometrids at lower elevations tended towards phylogenetic overdispersion, or have a random structure as is found here. The random structure we found may suggest that insect herbivore communities on diverse vegetation are not structured by competition as their resources are finely divided, with hundreds of potential host species coexisting in tropical vegetation. The lack of NRI and NTI overdispersion in lowland communities is consistent with the lack of strong correlation with biotic variables, including plant diversity and predation pressure (see section 4.3).

## 4.2 Community phylogenetic beta-diversity

We found low to moderate phylogenetic beta diversity among lowland communities and highly nested communities at higher elevation starting from 2700 m a.s.l. The relatively low beta diversity at lower elevations is consistent with earlier large-scale studies which found low beta-diversity in Lepidoptera and other herbivorous insects across lowland rainforests of New Guinea [77]. The widespread presence of large plant genera in lowlands facilitates this pattern, allowing Lepidoptera to expand their food niches across possible biotic and abiotic constraints [77]. Similarly on Mt. Wilhelm gradient, the natural range of the common plant genus *Ficus* persisted from lowland to 2700 m a.s.l [78,79] providing an opportunity for the moths to thrive to that limit while high elevation communities became distinctly dominated by generalist feeders [80].

As expected, phylogenetic beta diversity was mainly influenced by species turnover, as is the case in most tropical insect community studies [81]. However, nestedness became increasingly prominent at 2700 m a.s.l and above, where lowland communities were characterised more by the gradual loss of species with elevation than having a specific high-elevation fauna. In contrast, Szczepański et al. [82] found a strong turnover for *Cetiocyon* water beetles restricted to a particular elevation, with no shared lineages along the wider parts of the gradient. While lowland and mid-elevation sites are communities comprising, species restricted to certain elevational belts, highland communities (2700 m a.s.l and above) appear to be nested, with over 50% of species overlapping with lower elevation sites. This result is supported by the clusters of mean pairwise community distance separating highland communities (3200 and 3700 m a.s.l) from lowland ones which further formed two sub-ecologically similar clusters corresponding with the mid-elevation (1700–2700 m a.s.l) and lowland (200–1200 m a.s.l) communities.

## 4.3 Biotic and abiotic drivers

In many cases, the changes in species richness, abundance, and alpha diversity along elevational gradients are driven by biotic and abiotic factors, particularly temperature [31,32,56,83,84]. In this study however, we detected no strong relationships between Geometridae community phylogenetic structure and temperature, plant species richness, predation pressure or their combinations. This may suggest that the factors not considered in the study may be involved or that the relationships were not detectable at the spatial scale studied. However, there was a marginal linear effect of temperature on phylogenetic diversity (SES.PD) and phylogenetic diversity in turn was negatively strongly correlated with NTI. These results suggested that temperature is influencing the depauperated phylogenetic diversity towards higher elevations, selecting preadapted subfamily groups to thrive and co-occur there [21].

Other studies have also identified temperature as an important factor shaping the species communities and their phylogenetic structure along elevational gradients [31,32,76]. In ants for example, decreasing temperature and harsh environmental conditions selected fewer lineages to thrive at higher elevations [32]. Temperature also exhibited a strong correlation with NTI in ants along Neotropical gradients, illustrating the tendency for closely related lineages to co-occur with increasing elevation [21]. On Mt. Wilhelm, temperature has considerable effect on species diversity and composition of other taxonomic groups such as ants [36,85], *Ficus* plants [79,86], *Cetiocyon* beetles [82] and birds and butterflies [34,64] whereas its effect on phylogenetic diversities of these taxa remain unknown. Temperature not only influences single taxa but was shown to best predict species richness both in plants and animals at a community level for instance along 3.7 km of gradient in Mt. Kilimanjaro [87].

Apart from the effect of temperature, phylogenetic structure may be an indirect effect of temperature on plant diversity which in turn acts as a biotic filter for herbivore assemblages [31]. We found that plant species richness exhibited a marginal quadratic relationship with NTI. Because NTI measures pairwise phylogenetic distances at the tip label within the community [16,17], this result suggests that consistent with the effect of temperature, plant diversity is acting on species level OTUs at community level. The quadratic correlation also reflects their similar unimodal distribution with both diversities highest at mid elevation [29]. This is consistent with the notion that assemblages of tropical insects are strongly correlated with host plant phylogeny [88–91]. This result is supported by the significant effect of plant Sorensen dissimilarity on moth phylogenetic Sorensen dissimilarity suggesting that as plant communities become distinct with elevation, the moth communities that depend on them become more phylogenetically distinct following plant species composition [92]. This correlation however, is presumably weakened by the lack of geometrids feeding on some plant species any by some geometrid species feeding on non-woody plants or plant litter [58].

The abundance of predators had no impact on the phylogenetic diversity of the moths studied. The direct reason might be that the predation measured was indirect, not specifically focused on adult moths and there was also not-so-clear connection as to why it should be detectable into shifts in phylogenetic structure rather than in traits composition, activity patterns or other measures not considered here. Indirectly, this may also suggest that anti-predator evolutionary strategies are phylogenetically flexible [93]. For instance, it is possible that the moths have developed adaptive traits that provide an advantage to move between different elevations to find food or mates, therefore reducing the impact of predation [94], as detected here indicated by moderate beta diversity and turnover and random phylogenetic structure at low and mid elevations. According to Welti et al. [95], the effect of predation is distinct in a top-down control from predators influencing prey composition and more distinct in less complex systems [96] unlike the one we assessed. As an evolutionary advantage, moths with short, but

larger wing areas are commonly found at lower and mid-elevations, possibly as an adaptive strategy for flight agility [97], enabling them to move between adjacent elevational belts. On the other hand, moths with long, slender wings mostly characterised by subfamily Larentiinae [28,31,74] are restricted to higher elevations as a strategy for slow flying particularly where predation pressure is no longer a primary concern [80]. Based on an exclusion experiment [59], birds and bats however were shown to have significant top-down control on herbivore diversity on the same elevational gradient and this was probably influenced by some level of manipulation, such as disturbance, although in natural systems, predator prey interaction can be complex, especially in the tropical settings as studied here (e.g. no detection of distinct interactions).

## 5 Conclusion

The faunal composition and phylogenetic structure of geometrids along the slope of Mt. Wilhelm in New Guinea followed a typical pattern of a tropical mountain gradient with clustering towards higher elevations. Interestingly, we did not detect phylogenetic overdispersion, probably due to sampling limitations or caused by the factors not considered in the study. The turnover had a major contribution to the phylogenetic beta-diversity in lowland and mid-elevations while nestedness became prominent in higher elevations. Based on mean phylogenetic distances, the eight moth communities fell into three principal clusters, formed by lowland, mid-elevation, and highland communities even though no distinct natural ecotones exist except above 3200 m a.s.l [98] and the tendency of turnover changes according to these clusters. Our models for phylogenetic structure based on NRI, NTI, and SES.PD in relation to plant species richness, predator abundance, and temperature indicated loose relationships and suggested that the mechanisms beyond the tested variables may be at play and calls on further insect community phylogeny studies on the gradient to understand this.

## Supporting information

**S1 Fig. Scatter correlation matrix indicating relationship between elevation and the three phylogenetic matrices (NRI, NTI, SES.PD).** The x and y axis for each parameter is given according to the scale of measurements. Red lines are linear fit corresponding with the magnitude of correlations as shown by the correlation factors.
(TIF)

**S2 Fig.** Phylogenetic beta-diversity of Geometridae moth communities along Mt. Wilhelm gradient showing nestedness (A) and turnover (B). Both indices range from 0 to 1: 0 depicts species community having no nestedness (A) or unique communities (B) and the value of 1 depicts communities with high nestedness (A) and high shared species in turnover (B).
(TIF)

**S3 Fig. Phylogenetic beta-diversity of turnover and nestedness of Geometridae moths based on phylogenetic pairwise distance.** Turnover places more influence on the overall phylogenetic beta diversity.
(TIF)

**S4 Fig. Cluster dendrogram based on phylogenetic pairwise distances of 604 Geometridae species.** There are two principal clusters: one with highland communities formed by top two elevation sites (3200 and 3700 m a.s.l) and rest of the sites as lowland cluster further clustering to mid (2700, 2200 and 1700 m a.s.l) and lowland (1200, 700 and 200 m a.s.l) clusters relative to moth species genetic differences.
(TIF)

**S1 Table. Geometridae moth abundance, the number of morpho-species and the sequenced species occurrences per elevation.** The observed values of the predictors are given here, and their standardized z-score (SES) values are given in Table 2 in the main text. * Indicates the data used for this study.
(DOCX)

**S2 Table. The first 20/88 evolutionary models for 604 Geometridae moth species.** The best model is the one with lowest AIC score at the top as generated from IQtree model selection.
(DOCX)

**S3 Table. The *picante* run summary table for the phylogenetic structure in SES.PD, the mean pairwise distance (MPD) and its derived NRI and the mean nearest taxon distance (MNTD) and its derived NTI.** The moth communities with significant clustering are bolded. SES.PD, NRI and NTI are visualized as Fig 3 in the main text.
(DOCX)

**S4 Table. The table of second order polynomial regression of phylogenetic matrices to plant species richness, predator abundance and mean temperature as predictors for Geometridae moth phylogenetic structure.** The null models (all indicating significance) represented by intercept are highlighted grey while other significant models are bolded.
(DOCX)

**S1 File. The 1390 x elevation Geometridae species occurrence matrix used for the randomization analysis with genus level constrained phylogenetic tree object.**
(CSV)

**S2 File. The genus level constrained phylogenetic tree object of 604 Geometridae species used for the randomization analysis.**
(NEX)

**S3 File. The genus level constrained phylogenetic tree of 604 Geometridae species with branch support.**
(PDF)

**S4 File. Community pairwise phylogenetic distances among the eight communities of Geometridae moths along Mt. Wilhelm elevational gradient.**
(CSV)

# Acknowledgments

We thank landowners along the Mt. Wilhelm transect for having access to their forest, Katerina Sam for providing ecological data, Gabriel Saffa for statistical advice, staff, and students of New Guinea Binatang Research Centre for providing logistical support during field work, Niklas Wahlberg for providing 1206 published sequences used for the super-tree and Oscar Pérez Flores, Efrain de Jesus Carrillo, and Damian Villasenor Amador for help with some of the analyses and R-codes. DNA sequencing was provided by the Canadian Centre for DNA Barcoding, University of Guelph, through the iBOL project, funded by Genome Canada and others, and also supported by the National Museum of Natural History, Smithsonian Institution, USA. We thank Jeremy de Waard, Lauren Helgen, Margaret Rosati, Talitta Simoes, David Pollock, and Nicholas Silverson for their assistance with DNA barcodes. Axel Hausmann and Jeremy Holloway provided taxonomic advice. We thank our reviewers, Konrad Fiedler and Axel

Hausmann, for their attention to detail and for providing critical comments, which improved our manuscript.

## Author Contributions

**Conceptualization:** Sentiko Ibalim, Pagi S. Toko, Simon T. Segar, Scott E. Miller, Vojtech Novotny.

**Data curation:** Sentiko Ibalim, Pagi S. Toko, Bonny Koane, Milan Janda.

**Funding acquisition:** Scott E. Miller, Vojtech Novotny.

**Methodology:** Sentiko Ibalim, Pagi S. Toko, Simon T. Segar, Katayo Sagata, Bonny Koane.

**Project administration:** Sentiko Ibalim, Scott E. Miller, Vojtech Novotny, Milan Janda.

**Supervision:** Milan Janda.

**Writing – original draft:** Sentiko Ibalim, Simon T. Segar, Katayo Sagata, Vojtech Novotny, Milan Janda.

**Writing – review & editing:** Sentiko Ibalim, Pagi S. Toko, Simon T. Segar, Scott E. Miller, Vojtech Novotny, Milan Janda.

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
