## [Decision Letter · Decision Letter 0]

3 Jul 2024

PONE-D-24-23196Phylogenetic structure of moth communities (Geometridae, Lepidoptera) along a complete rainforest elevational gradient in Papua New Guinea.PLOS ONE

Dear Dr. Ibalim,

Thank you for submitting your manuscript to PLOS ONE. After careful consideration, we feel that it has merit but does not fully meet PLOS ONE’s publication criteria as it currently stands. Therefore, we invite you to submit a revised version of the manuscript that addresses the points raised during the review process.

We look forward to receiving your revised manuscript.

Kind regards,

Mostafa Ghafouri Moghaddam, Ph.D

Academic Editor

PLOS ONE

Journal Requirements:

   "The study was supported by the Praemium Academiae to VN Czech Science Foundation (23-07776S), The European Research Council (669609) and GAJU n. 014/2022/P supporting SI and by CONAHCYT program Investigadores por Mexico, project No. 338. 338 to SEM. "

5. We note that Figure 1 in your submission contain map/satellite images which may be copyrighted. All PLOS content is published under the Creative Commons Attribution License (CC BY 4.0), which means that the manuscript, images, and Supporting Information files will be freely available online, and any third party is permitted to access, download, copy, distribute, and use these materials in any way, even commercially, with proper attribution. For these reasons, we cannot publish previously copyrighted maps or satellite images created using proprietary data, such as Google software (Google Maps, Street View, and Earth). For more information, see our copyright guidelines: http://journals.plos.org/plosone/s/licenses-and-copyright.

Additional Editor Comments:

Dear Authors,

I have received the reviewers' comments and would like to share the details with you. Based on the reviewers' feedback and my own assessment, my decision is "Minor Revision." Please carefully address all comments and make the necessary revisions accordingly.

Ensure that you improve both the technical and linguistic aspects of your manuscript. Once you have made the revisions, please send the updated manuscript back to me at your earliest convenience.

Congratulations on your excellent work in advance. I am confident that with these minor revisions, your manuscript will be even more impactful.

Looking forward to hearing from you soon.

All the best,

Mostafa Ghafouri Moghaddam, PhD

Chulalongkorn University, Bangkok, Thailand

Reviewers' comments:

Reviewer's Responses to Questions

**Comments to the Author**

1. Is the manuscript technically sound, and do the data support the conclusions?

Reviewer #1: Yes

Reviewer #2: Yes

2. Has the statistical analysis been performed appropriately and rigorously? 

Reviewer #1: Yes

Reviewer #2: Yes

3. Have the authors made all data underlying the findings in their manuscript fully available?

Reviewer #1: Yes

Reviewer #2: Yes

4. Is the manuscript presented in an intelligible fashion and written in standard English?

Reviewer #1: Yes

Reviewer #2: Yes

5. Review Comments to the Author

Reviewer #1: This paper describes elevational patterns in phylogenetic diversity within a speciose clade of moths across a continuous, complete forested gradient in the mountains of Papua New Guinea. The paper is well written, addresses all the relevant literature and is based on extensive samples that have been worked out through a combination of morphological and molecular taxonomy. Also the data analysis uses up-to-date methodology. The main findings are (a) that phylogenetic diversity shows a clear mid-elevation peak and (b) that none of 3 tested and plausible environmental drivers (temperature, species richness of woody vegetation, and abundance of potential predators) showed any clear correlation with moth diversity. Moreover, at high elevations geometrid assemblages tend to be phylogenetically clustered, whereas this was not the case at lower sites; and there was no evidence for phylogenetic overdispersion.

The main weakness of the study is the low number of spatial replicates, which precludes any more sophisticated analyses. But given the many obstacles to be surmounted when assembling such a wonderful data set, I see that as a minor point – evidently these data will serve as a benchmark and base-line for decades to come.

I have only a handful of suggestions when revising the paper.

L 35: typo, should read Sorensen

L 228: perhaps better “quantifying potential resources”, since I suspect host plant relationships of most of these PNG moths remain undocumented thus far.

L 233: while I accept that focussing on woody plants was the only feasible way to assess vegetation diversity in such a diverse and largely inaccessible rainforest, it should be noted somewhere in the paper that this approach ignores potentially relevant larval food resources such as understory herbs, epiphytes, and perhaps in some geometrid species also litter, lichens, fungi and alike; see for example https://doi.org/10.1017/S0266467415000243.

L 257: you mainly used phylogenetic dissimilarity matrices to partition them into turnover and nestedness components and to check for the strength of differences relative to elevation. This is all fine. However, I suggest you could also attempt to relate, using (Spearman?) matrix correlations, the spatial pattern of community differences (Phylo-Sorensen matrix) with matrices that describe the (Euclidean) distance between pairs of sites in terms of temperature, woody plant richness, and predator abundance. While you did not observe any strong relationships with regard to (phylogenetic) alpha diversity of geometrids and these 3 potential drivers, perhaps you see more pattern if you use beta diversity (more precisely: differentiation diversity) as a response variable here?

Reviewer #2: The authors present a fine paper based on community phylogenetics of Geometridae (looper moths) to elucidate the community assembly mechanisms collected along an elevational rainforest gradient (remarkably comprising an altitudinal range of 3500 m!) on Mount Wilhelm in Papua New Guinea. A constrained phylogeny was based on COI barcodes for 604 species. The community patterns of moths are compared with several biotic and abiotic parameters, like plant species richness, predator abundance and mean temperature.

The phylogenetic analysis was performed in an appropriate way and the presented conclusions sound reliable and well-substantiated. The discussion of the results is very accurate and detailed. The figures are well prepared and the supporting info well prepared with scientific names almost without misspellings.

I recommend to accept this paper in the present form, after correcting the very few details as following.

A few details:

Line 29: a.s.l -> a.s.l. (also in line 121, 123 a.s.o.) (alternatively „AMSL“?)

Line 46: altitudinal -> Altitudinal

Line 51: Does it really make sense to write „1.0“ here? I would recommend to change it to „1.“

Line 117: Does it really make sense to write „2.0“ here? I would recommend to change it to „2.“ (a.s.o.)

Line 293: please take away the dot at the end (you don’t have it in the other headlines)

Line 297: 3700m -> 3700 m (insert spaces also in lines 330, 333, 371, 373, 374, 375, 416 etc.)

Line 411: ( see -> (see

Line 422: 2700 -> 2700 m

Suppl. File 1, line 89: _cliptopera -> _Ecliptopera

Suppl. File 1, lines 330, 331: _comostola -> _Comostola

6. PLOS authors have the option to publish the peer review history of their article (what does this mean?). If published, this will include your full peer review and any attached files.

Reviewer #1: **Yes: **Konrad Fiedler

Reviewer #2: **Yes: **Axel Hausmann

---

## [Author Response · Author response to Decision Letter 0]

24 Jul 2024

Dear reviewers, we have completed the revisions for our revised manuscript # PONE-D-24-23196 and address each comments point by point. We hope the revised manuscript now satisfies your expectations and meet the journal's requirements.

---

## [Editor Report · Decision Letter 1]

30 Jul 2024

Phylogenetic structure of moth communities (Geometridae, Lepidoptera) along a complete rainforest elevational gradient in Papua New Guinea.

PONE-D-24-23196R1

Dear Dr. Sentiko Ibalim,

We’re pleased to inform you that your manuscript has been judged scientifically suitable for publication and will be formally accepted for publication once it meets all outstanding technical requirements.

Kind regards,

Mostafa Ghafouri Moghaddam, Ph.D

Academic Editor

PLOS ONE

---

## [Editor Report · Acceptance letter]

2 Aug 2024

PONE-D-24-23196R1 

PLOS ONE

Dear Dr. Ibalim, 

I'm pleased to inform you that your manuscript has been deemed suitable for publication in PLOS ONE. Congratulations! Your manuscript is now being handed over to our production team.

Kind regards, 

on behalf of

Dr Mostafa Ghafouri Moghaddam 

Academic Editor

PLOS ONE